



# Improved definition of prior uncertainties in CO₂ and CO fossil fuel fluxes and the impact on a multi-species inversion with GEOS-Chem (v12.5)

Ingrid Super[1], Tia Scarpelli[2], Arjan Droste[1,3], Paul I. Palmer[2,4]

[1]Department of Climate, Air and Sustainability, TNO, P.O. Box 80015, 3508 TA Utrecht, the Netherlands
5   [2]School of GeoSciences, University of Edinburgh, UK
[3]Department of Water Management, Water Resources section. Faculty of Civil Engineering & Geosciences, Delft Technical University, the Netherlands
[4] National Centre for Earth Observation, University of Edinburgh, UK

*Correspondence to*: Ingrid Super (ingrid.super@tno.nl)





**Abstract.** Monitoring, reporting and verification frameworks for greenhouse gas emissions are being developed by countries across the world to keep track of progress towards national emission reduction targets. Data assimilation plays an important role in monitoring frameworks, combining different sources of information to get the best possible estimate of fossil fuel emissions and as a consequence better estimates for fluxes from the natural biosphere. Robust estimates for fossil fuel emissions rely on accurate estimates of uncertainties corresponding to the different pieces of information. We describe prior uncertainties in $CO_2$ and CO fossil fuel fluxes, with special attention paid to spatial error correlations and the covariance structure between $CO_2$ and CO. This represents the first time that the prior uncertainties in $CO_2$ and the important co-emitted trace gas CO are defined consistently, including error correlations, which allows us to make use of the synergy between the two trace gases to better constrain $CO_2$ fossil fuel fluxes. The $CO:CO_2$ error correlations differ per sector, depending on the diversity of sub-processes occurring within a sector, and also show a large range in values between pixels for the same sector. For example, for other stationary combustion the pixel correlation values range from 0.1 to 1.0, whereas for road transport the correlation is mostly larger than 0.6. We illustrate the added value of our prior uncertainty definition using closed-loop numerical experiments over mainland Europe and the UK, which isolate the influence of using error correlations between $CO_2$ and CO and the influence of prescribing more detailed information about prior emission uncertainties. We find that using our realistic prior uncertainty definition helps our data assimilation system to differentiate more easily between $CO_2$ fluxes from biogenic and fossil fuel sources. Using the improved prior emission uncertainties we find fewer geographic regions with significant changes from the prior than using the default prior uncertainties, but they almost consistently move closer to the prescribed true values, in contrast to the default prior uncertainties. We also find that using CO provides additional information on $CO_2$ fossil fuel fluxes, but only if the $CO:CO_2$ error covariance structure is defined realistically. Using the default prior uncertainties, the $CO_2$ fossil fuel fluxes move farther away from the truth for many geographical regions. With the default uncertainties the maximum deviation of fossil fuel $CO_2$ from the prescribed truth is about 7 % in both the prior and posterior result. With the advanced uncertainties this is reduced to 3 % in the posterior.



## 1 Introduction

With the signing of the Paris Agreement, 195 nations have committed themselves to reducing their greenhouse gas (GHG) emissions. This calls for active monitoring of emissions and emission trends to ensure climate plans are being met. Work is currently ongoing to build a GHG Monitoring, Reporting and Verification framework (MRV), which will track and verify emissions of the major GHGs using a multi-tiered observing system. The MRV will support the 5-yearly global stocktake (Balsamo et al., 2021; Janssens-Maenhout et al., 2020; Petrescu et al., 2021), and increase the understanding of emission

landscapes and the associated dominant source sectors that is necessary to develop effective nationwide emission mitigation strategies to support national determined contributions.

An important aspect of the MRV is combining different types of data, e.g., spatially disaggregated bottom-up inventories, atmospheric data, and near-real-time weather and economic data, to obtain the best possible estimate of the national fossil fuel GHG emissions. This is often done through data assimilation (or inverse modelling), which is a rigorous mathematical

framework to combine all these pieces of information (Lauvaux et al., 2016; Pillai et al., 2016; Staufer et al., 2016; Wu et al., 2018). GHG data assimilation uses state-of-the-art atmospheric transport models, prior information on GHG sources and sinks, observational data and the uncertainties in each of these data sources. The uncertainties determine how much confidence is input in each of the components and thus how much information is taken from them, but some of these uncertainties, like model transport uncertainties, are notoriously difficult to estimate. A limiting factor is often the lack of sufficient high-quality

observations. Although a relatively dense GHG monitoring network exists in some countries, e.g., the UK, mainland Europe and North America, many regions only have very sparse observations. Satellite data can significantly increase that coverage and have proven useful in specific cases. For an MRV targeting combustion $CO_2$, one major limitation of satellite data is that observations are atmospheric columns that include a large background concentration (Broquet et al., 2018; Chevallier et al., 2022; Palmer et al., 2008; Reuter et al., 2019).

One way to isolate the signal from combustion emissions is by exploiting the synergy between $CO_2$ and co-emitted species, such as CO and $NO_x$, which share the same combustion sources. Many countries have an air quality monitoring network and many air pollutants are being observed from space (e.g., CO, $NO_2$), with the advantage of having relatively short e-folding lifetimes (< few months) and consequently having a smaller background contribution. Several studies have explored the correlation between $CO_2$ and co-emitted species and the additional constraint co-emitted species provide on $CO_2$ emissions,

both with in-situ and satellite data (Boschetti et al., 2018; Brioude et al., 2013; Palmer et al., 2022; Reuter et al., 2019; Silva et al., 2013; Turnbull et al., 2006; Yang et al., 2023). Co-emitted species have been used to separate fossil fuel $CO_2$ from biogenic $CO_2$ signals (Oney et al., 2017; Suntharalingam et al., 2004; Vardag et al., 2015), to estimate $CO_2$ emissions without $CO_2$ observations (Konovalov et al., 2016; Liu et al., 2020; Lopez et al., 2013), and to allocate $CO_2$ signals to specific emission sectors (Nathan et al., 2018; Super et al., 2020b; Turnbull et al., 2015). The latter makes use of the sector-specific emission

ratio of $CO_2$ and co-emitted species.



Although there is promise in this multi-species approach, the emission ratios are uncertain, dynamic in space and time (Ammoura et al., 2016; Liñán-Abanto et al., 2021; Super et al., 2017; Wu et al., 2022), and may even depend on human behaviour or meteorological conditions (Ammoura et al., 2014; Hall et al., 2020). The objective of data assimilation is to reduce the mismatch between posterior estimates and observations, so that co-emitted species are only useful for informing

$CO_2$ emissions if the uncertainties for the $CO_2$ emission estimates are larger than uncertainties associated with the observed ratios between $CO_2$ and co-emitted species. Therefore, an important role is laid out for accurately assessing the uncertainties of prior emissions, and the definition of error correlations, which is a complex task. Gridded prior emissions are based on several data sources and therefore include uncertainties in activity data, emission factors, and the spatial and temporal patterns. Some of these uncertainties might also be correlated, e.g., between regions and/or trace gases. Error correlations describe the

synergy in emission uncertainties and can increase the amount of information gained from the same input data. One example is that gridded uncertainties are not independent from uncertainties in nearby grid cells.

In the context of simultaneously optimizing $CO_2$ and CO emission estimates, the most uncertain parameter in the calculation of CO emissions is the emission factor for which the error is not correlated to the error in the $CO_2$ emission factor. This limits the use of CO in constraining $CO_2$ at the national scale (Palmer et al., 2006). However, CO and $CO_2$ emissions are correlated

through fossil fuel combustion activity, which determines to a large extent the spatial patterns of the emissions. In practice, the spatial distribution for CO and $CO_2$ emission estimates is often based on the same spatial data. Therefore, gridded CO and $CO_2$ emission estimates show a much stronger error correlation than the national emissions. So that the error in $CO_2$ emissions in one grid cell is likely to be similar to the error in CO emissions in the same grid cell, because the errors are caused by the assumed shared activity. Hence, by quantifying the gridded error correlations we can make better use of the information from

CO to constrain $CO_2$.

Few studies have tried to estimate gridded emission uncertainties (Gately and Hutyra, 2017; Hogue et al., 2016; Hutchins et al., 2017; Oda et al., 2019), but only for $CO_2$. These studies mostly compare different emission datasets, which likely underestimate the uncertainties when the inventories use similar underlying data. Super et al. (2020a) provided a bottom-up uncertainty estimate of gridded emissions for $CO_2$ and CO, for which an emission inventory was used that has a consistent

methodology for $CO_2$ and co-emitted species. This increases the use of error correlations between $CO_2$ and co-emitted species. In this previous work spatial errors were treated as independent, and no spatial correlations were considered. Also, the error correlation between $CO_2$ and co-emitted species was not examined.

Here we describe an effort to build a consistent set of prior emission uncertainties for $CO_2$ and co-emitted species (CO), building further on the work done by Super et al. (2020a). This paper starts with a description of the data (Sect. 2.1) and

methodology (Sect. 2.2) used to develop a more detailed prior uncertainty definition, including spatial error correlation lengths and the error correlation between $CO_2$ and CO. The results are shown in Sect. 3.1. To illustrate the added value of well-defined prior uncertainty information we perform closed-loop numerical experiments, as explained in Sect. 2.3. We show the results for $CO_2$-only (Sect. 3.2) and multi-species inversions (Sect. 3.3).



## 2. Methods

This section starts with a description of the data used to make a detailed prior uncertainty definition for gridded emissions, including the prior emission inventory. For this we separate uncertainties in activity data/emission factors and in the spatial patterns. Next, we describe the methodology used to estimate spatial error correlations and the error correlation between CO and $CO_2$. Additionally, we discuss how all uncertainties are combined in one product that can be used in data assimilation studies. Finally, we describe the set-up of the inversions, including a description of the models, state vector, input data and the

different experiments.

    In this work we use the words 'uncertainties' and 'errors', which have a slightly different meaning. We do not know the exact errors in our data, so we talk about uncertainties to define how reliable our data are. When referring to correlations we use the term 'error'. For example, if errors between neighbouring grid cells are positively correlated it means that if we overestimate the value in a one grid cell we are likely to do the same in the other grid cell. In this case we are talking about actual errors,

which we cannot define, but we know are correlated. It is not the uncertainty that is correlated. For the same reason we use the term 'error covariance matrix'.

### 2.1 Prior data

### 2.1.1 European emission dataset

    The European prior emission dataset used as a basis for this work is the TNO-GHGco-v4 for 2018 with a spatial resolution of

0.1° by 0.05°. This dataset provides a unique set of consistent emissions for a range of GHGs and co-emitted species (Fig. 1), which allows us to study the impact of error correlations between these species on data assimilation studies.

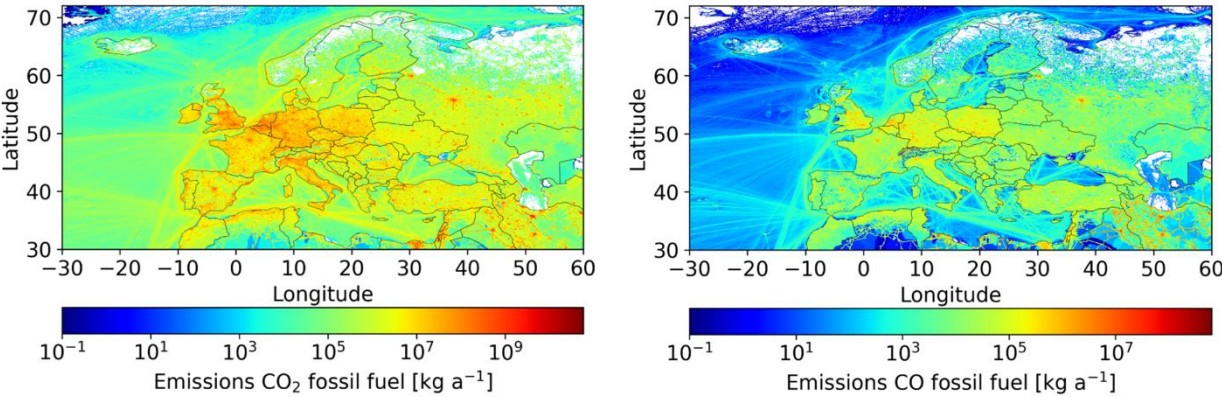

**Figure 1: TNO-GHGco-v4 emission maps of $CO_2$ and CO for 2018.**

    The TNO-GHGco-v4 dataset is similar to the CAMS-REG emission inventory (Kuenen et al., 2022), except that point sources

are placed at their exact location instead of taken up by the grid cells. It is compiled from emission reports delivered to CEIP (Data reported by Parties under LRTAP Convention, 2022) and the UNFCCC (National Inventory Submissions 2020, 2022) by individual countries. The reports contain emissions for a long list of sub-sectors and fuels. In the final dataset these





emissions are aggregated into 12 sectors (GNFR (Gridded Nomenclature For Reporting) categorisation, see Table 1). For countries that do not report their emissions, other emission datasets are used for gap filling that are only available at the GNFR

level. For the uncertainty estimates we work with the detailed reported emission data. In the final product we aggregate to GNFR sectors A, B, C, F and G, and a sixth group for the remaining minor sectors.

The country-level emissions are spatially downscaled to 0.1° by 0.05° resolution using proxy maps (Kuenen et al., 2022). The proxy maps describe the fraction of the country-level emissions for a particular sub-sector that is assigned to one grid cell, such that the fractions sum up to 1 for each country-sector combination. Some proxy maps are used for multiple sub-sectors.

For some countries the spatial proxies are not available or replaced with other datasets.

For GNFR G a different approach is used because most of the emissions in this sector occur on international waters and are therefore not reported by countries. All shipping emissions are therefore taken directly from the STEAM model (Jalkanen et al., 2012; Johansson et al., 2017), that provides gridded emissions using AIS (Automatic Identification System) data and vessel characteristics.

**Table 1: Overview of aggregated emission categories in the European emission data (GNFR), including IPCC-based relative uncertainties (95 % confidence interval (CI)) in activity data (AD) and in $CO_2$ emission factor (EF) per GNFR sector are given, which are used for countries without reporting.**

| GNFR category | GNFR category name | AD rel. unc. [%] | EF rel. unc. [%] |
|---|---|---|---|
| **A** | A_PublicPower | 2.0 | 4.9 |
| **B** | B_Industry | 3.0 | 4.9 |
| **C** | C_OtherStationaryComb | 15.0 | 4.9 |
| **D** | D_Fugitives | 5.0 | 75.0 |
| **E** | E_Solvents | | |
| **F** | F_RoadTransport | 5.0 | 5.0 |
| **G** | G_Shipping | 5.0 | 1.5 |
| **H** | H_Aviation | 50.0 | 5.0 |
| **I** | I_OffRoad | 50.0 | 2.0 |
| **J** | J_Waste | 13.5 | 7.1 |
| **K** | K_AgriLivestock | | |
| **L** | L_AgriOther | 20.0 | 20.0 |

### 2.1.2 Country-level emission uncertainties

In the emission reporting, over 250 different sector-fuel combinations are differentiated. We make a pre-selection of these by

ordering them based on their total emissions for the whole European domain. Then we select the most important sector-fuel combinations until we include at least 95 % of the emissions for all species. We combine the selections for all species, so they are all the same, and we end up with 90 sector-fuel combinations that describe 96 % of $CO_2$ emissions, 98 % of CO emissions,



and 97 % of NO$_x$ emissions. For the selected sector-fuel combinations we gather uncertainty data. All other sector-fuel combinations get an uncertainty of zero. A summary of the country-level uncertainties is provided in Table S1.

Most countries in the European domain are Annex-I countries, which report their GHG emissions annually to the UNFCCC following standardized reporting guidelines. Most countries also include an uncertainty estimate in their National Inventory Report (NIR), with separate uncertainty estimates for activity data (AD) and the emission factor (EF), which form the starting point for our work. For CO, such reported uncertainties are not available. Because CO shares the AD with CO$_2$, we use the CO$_2$-based reported country-level uncertainties. For the EF uncertainty in CO we use global EF uncertainty data per sector-

fuel combination from the most recent EMEP guidebook (European Environment Agency, 2019). These uncertainties are applied to all countries, irrespective of whether emissions are reported or taken from another emission dataset. The gap filling procedure is explained in the SI.

   For countries in the emission inventory domain that do not report GHG emissions to UNFCCC we estimate the uncertainties at the GNFR level from the IPCC guidelines (Eggleston et al., 2006). Since the emission factor for CO$_2$ depends only on the

fuel type, and not on the combustion technology, the uncertainty ranges are generic and consistent across sub-sectors. Where needed we have averaged uncertainty ranges, selected the largest uncertainties from a given range, or made an estimate based on the reported uncertainties from Annex-I countries. For GNFR G a separate estimate has been made for the activity, based on a comparison of the STEAM model predictions with fuel reporting (J.-P. Jalkanen, personal communication, August 25,2022), and the emission factors (Grigoriadis et al., 2021). This results in the uncertainties given in Table 1. Note that GNFR

sectors E and K are missing because they are irrelevant for CO$_2$.

### 2.1.3 Emission proxy map uncertainties

   The spatial uncertainties in the emissions are caused partly by the discrete nature of the grid, but more importantly by uncertainties in the proxy maps used for downscaling the national emissions. There are different sources of uncertainty in the proxy maps. The three main ones are: the value of each pixel, e.g., the population density which might be lower or higher than

in reality; the quality of the proxy, e.g., whether there are cells missing that contain an activity (or v.v.); and the representativeness of the proxy for the activity causing the emissions, e.g., the ability of a population density map to reflect residential combustion emissions.

   We include detailed spatial uncertainties for two GNFR sectors (road transport (GNFR F) and other stationary combustion (GNFR C)), which are important contributors to CO and CO$_2$ emissions and have the strongest CO:CO$_2$ error correlation.

These sectors each consist of several sub-sectors that are downscaled with different proxy maps. By starting at the sub-sector level each grid cell will receive a unique uncertainty at the GNFR level, depending on the mix of sub-sectors. An overview of the proxy maps for these two GNFR sectors is given in Table 2. Note that spatial uncertainties are only included for those countries for which emissions are downscaled using these proxies.

   We start with the accuracy of the pixel value. The proxies for road transport are based on OpenTransportMap (Jedlička et al.,

2016), which combines the OpenStreetMap (OSM) road network with traffic volume from traffic simulation models. OSM is




community-based and is not always complete or accurate. Yet, the main source of uncertainty is from the underlying traffic simulation models. A wide range of models exists, each having their own strengths and weaknesses. Some guidance on their accuracy is given by Gao et al. (2010), who calculated an average RMSE of 31 % in traffic volume for two traffic models (MATSim and EMME/2). For the VISUM model a similar mean relative error of 30 % was found (Raney et al., 2003). These
studies therefore indicate a 95 % CI of about 60 % (about two times the RMSE). However, both studies are performed at a very high resolution (street links), whereas our resolution is much coarser (~ 6 km grid cells). Therefore, the uncertainty in our proxy map is probably smaller and we set the 95 % CI to 30 %. The population density is based on LandScan (Bright et al., 2016). This product describes the ambient population, which includes working and travelling population, by taking a 24-hour average. Archila Bustos et al. (2020) compared LandScan to population data from the Swedish Statistical Bureau and found
an average RMSE of 9 %, with larger errors for sparsely populated areas. This suggests the uncertainty distribution is skewed, as was also shown for Poland (Calka and Bielecka, 2019). Here, we assume the RMSE to be based on a large-enough population, since the largest absolute errors occur in densely populated areas, and estimate that the 95 % CI is more or less equal to two times the RMSE. Finally, the wood use proxy is based on population density and the proximity to wood/forests (Kuenen et al., 2022). The uncertainty is expected to be large, as the locations where residential wood burning takes place are
relatively unknown. For example, Grythe et al. (2019) demonstrated large differences in PM emissions from residential wood combustion between different datasets, even aggregated over large urban domains. We set the 95 % CI to 50 %.

**Table 2: Overview of proxy maps used for downscaling GNFR C and F, their 95 % CI and correlation length.**

| Proxy map | Uncertainty (95 % CI) | Correlation length (km) |
|---|---|---|
| RoadTransport_Urban_PC | 0.6 | 15 |
| RoadTransport_Urban_Mopeds | 0.6 | 15 |
| RoadTransport_Urban_Motorcycles | 0.6 | 15 |
| RoadTransport_Urban_HDV | 0.6 | 15 |
| RoadTransport_Urban_LDV | 0.6 | 15 |
| RoadTransport_Urban_Buses | 0.6 | 15 |
| RoadTransport_Highway_HDV | 0.6 | 28 |
| RoadTransport_Highway_LDV | 0.6 | 28 |
| RoadTransport_Highway_Buses | 0.6 | 28 |
| RoadTransport_Highway_PC | 0.6 | 28 |
| RoadTransport_Highway_Motorcycles | 0.6 | 28 |
| RoadTransport_Highway_Mopeds | 0.6 | 28 |
| RoadTransport_Rural_Buses | 0.6 | 21 |
| RoadTransport_Rural_LDV | 0.6 | 21 |
| RoadTransport_Rural_HDV | 0.6 | 21 |





| | | |
|---|---|---|
| **RoadTransport_Rural_Motorcycles** | 0.6 | 21 |
| **RoadTransport_Rural_Mopeds** | 0.6 | 21 |
| **RoadTransport_Rural_PC** | 0.6 | 21 |
| **Population_total_2015** | 0.36 | 23 |
| **Population_rural_2015** | 0.36 | 23 |
| **Population_urban_2015** | 0.36 | 23 |
| **Wood_use_2014** | 1.0 | 26 |

The second source of uncertainty is the proxy quality, which is a difficult uncertainties with which to work. There is no way
to correct a grid cell that falsely lacks activity, as scaling a value of zero always returns zero. This is mainly an issue for categorical proxies. Similarly, if the location of a point source is incorrect it is difficult to estimate where it should be instead. Since we cannot reliably compensate for this uncertainty, we have chosen not to account for it, while acknowledging it as a local source of uncertainty in the location of emissions.

Finally, the representativeness error behaves differently from the uncertainty in the pixel values. Aside from adding an
uncertainties to each pixel, it also causes errors to be correlated between pixels that have similar characteristics. For example, the heating demand for residential buildings depends on the population density. People that live closer together, e.g., in high-rise buildings, generally need less heating per person. This means that the heating emissions are not linearly related to population. If we make an error in describing this relationship it will affect pixels with similar characteristics in a similar fashion and hence errors are spatially correlated. We double the pixel value uncertainty to include the representativeness error
(Table 2). Moreover, we consider its impact on the error correlation, which is discussed in Sect. 2.2.1.

The other sectors receive a fixed uncertainty (95 % CI) for all grid cells, based on expert judgement. The sectors public power and industry contain point sources, for which the locational error can be large (Hogue et al., 2016). However, for the TNO-GHGco-v4 emission inventory locations have been thoroughly checked and we assume no spatial uncertainty. The remainder (non-point sources) receives an uncertainty of 200 %. For sea shipping the spatial patterns are relatively well-known using
AIS data (J.-P. Jalkanen, personal communication, September 15,2022) and we assume no spatial uncertainty. However, the AIS coverage on inland waterways is limited and therefore we set the uncertainty similar to that of the road transport sector (60 %). The other sectors are minor, but have a large spatial uncertainty. Since they are grouped some errors may cancel each other out and we assume an overall uncertainty of 200 %.

## 2.2 Prior emission uncertainties

In this section we describe how the prior emission uncertainties were calculated. An overview of all the steps is given in Fig. 2. Details are described below.





**Figure 2: Diagram of all the steps taken to calculate prior emission uncertainties and covariances.**

### 2.2.1 Spatial error correlation length

The representativeness error in a proxy map causes errors to be spatially correlated. We define the error correlation length as the maximum distance at which two grid cells are still correlated. This length scale is estimated by fitting spherical and exponential semi-variograms for each proxy map in Table 2 per country, using the fit.variogram function from the *gstat*





geostatistical package in the R software (Pebesma and Wesseling, 1998), and taking the range parameter. We set the limits of the considered distance between 6 km (original grid spacing) and 120 km.

The fitting procedure optimizes the model parameters to provide the best fit to the data and shows only small differences between the spherical and exponential models. We do this twice: once without setting an initial sill (the semi-variance at distance zero), and once by setting it to zero. This is to ensure that the resulting range values are not just the cause of the initial values set in the model. This results in two range values per country per proxy map and we pick the value that is within our set boundary, or the average of the two if both values are within this range. We can only use one correlation length for the

whole domain to avoid irregularities near country borders and therefore we take the median of all country-specific ranges. The results are illustrated in Sect. 3.1.

For the industry and public power, we set the error correlation length to zero since they are dominated by point sources. For shipping we estimate an error correlation length of 100 km, which is larger than for road transport given that for ships it is more difficult to take a turn.

**2.2.2 Error correlation between CO and $CO_2$ emissions**

The proxy maps used for the spatial downscaling are the same for all trace gases in the emission inventory, i.e., the $CO_2$ emissions of sub-sector X are downscaled with the same proxy map as the CO emissions of sub-sector X. This means that at the sub-sector level the spatial errors are strongly correlated between all trace gases. Because the mix of sub-sectors within an aggregated sector can be different for CO than for $CO_2$, the error correlation is reduced. Therefore, we define a predictor to

estimate the error correlation between CO and $CO_2$ in each grid cell for other stationary combustion and road transport, validated against a Monte-Carlo based correlation coefficient for seven countries that reflect relevant variations in the domain (Czech Republic, Germany, France, UK, Italy, Netherlands, and Sweden). This predictor allows us to calculate the error correlation for all grid cells without having to do an expensive Monte-Carlo simulation.

The predictor ($P$) is based on the CO and $CO_2$ emissions per grid cell $c$ per proxy map $m$ for the selected GNFR sector and the

uncertainties (standard deviation σ) in the proxy maps. In Eq. (1) STD is the standard deviation of the emissions per grid cell, proxy map and trace gas. When the relative contribution of each fraction map differs strongly between CO and $CO_2$, the correlation is weaker, which is expressed in the vector WD (Eq. (2)). The larger the number of proxy maps used for downscaling emissions from a particular sector, the stronger the correlation generally is. This is due to the damping effect on outliers. This results in the following set of equations:

$$STD_{c,m,g} = f_{c,m} \cdot E_{m,g} \cdot \sigma_m, \tag{1}$$

$$WD = \left| \frac{STD_{c,m,CO2}}{\sum (W_{c,m,CO2})/n} - \frac{STD_{c,m,CO}}{\sum (W_{c,m,CO})/n} \right|, \tag{2}$$

$$P_C = \frac{stdev(WD)}{n} \quad \text{and} \quad P_F = \frac{\max (WD)}{n}, \tag{3}$$

where $g$ is either one of the two trace gases CO or $CO_2$, $f$ is the fraction of a proxy map in a grid cell, and $n$ is the number of proxy maps with a contribution in a grid cell. Note that the predictor is slightly different for GNFR C and GNFR F. We define



a relationship between the predictor and the Monte-Carlo based correlation coefficient to calculate the $CO:CO_2$ error correlation per grid cell based on the predictor. Results are shown in Sect. 3.1 and the SI.

For the other sectors we estimate a fixed value for all grid cells. For public power and shipping the correlation is likely to be very strong, since there is little variation in sub-sector activities, so we set the error correlation to 0.95. For industry the correlation is much smaller due to different sub-processes taking place and we set the error correlation to 0.5.

### 2.2.3 Uncertainty propagation

We have now gathered all relevant information on uncertainties, which need to be propagated to match the level of detail of the prior emission dataset.

The country-level uncertainties represent a 95 % CI (normalized, so in unitless numbers), which is either given as one value or as a lower and upper value. For the latter, when the lower and upper values show less than 5 % difference we use a Gaussian

uncertainty distribution, otherwise we use a lognormal uncertainty distribution. For CO the uncertainty distribution is often lognormal. When the reported standard deviation exceeds 30 %, we also use a lognormal uncertainty distribution to avoid getting negative values. We use uncertainty propagation to estimate the uncertainty in the emissions from the standard deviations σ in AD and EF:

$$\frac{\sigma_E}{E} = \sqrt{\left(\frac{\sigma_{AD}}{AD}\right)^2 + \left(\frac{\sigma_{EF}}{EF}\right)^2}. \tag{4}$$

To examine the importance of error correlations in AD and EF we performed a sensitivity analysis on the European emissions (see SI). We found that including error correlations in AD and EF has limited importance and henceforth we ignore these correlations.

We approximate the standard deviation of a Gaussian distribution belonging to a lognormal distribution with:

$$\frac{\sigma_X}{X} = \frac{(\ln(lim_{upper}) - \ln(lim_{lower}))}{4}, \tag{5}$$

where $lim_{upper}$ is the 97.5 percentile and $lim_{lower}$ the 2.5 percentile of the lognormal distribution. Note that the combination of a Gaussian and lognormal function does not result in a lognormal function, because the result can be negative. However, here we assume that the combined distribution is lognormal, because the Gaussian uncertainty is often relatively small compared to the lognormal uncertainty.

The sub-sector level emission uncertainty estimates are propagated to get an uncertainty estimate at the GNFR level:

$$\sigma_{E,agg} = \sqrt{\sum_{i=m}^{n} \sigma_{E,sub,m}^2}, \tag{6}$$

where the subscript 'agg' refers to the aggregated emissions and uncertainties and the subscript 'sub' refers to the sub-sectors part of that aggregated sector. To use Eq. (6) we need the emission budgets, because it uses actual standard deviations instead of normalized ones.

These simple uncertainty propagation functions work well under specific circumstances. When uncertainties follow a non-

Gaussian distribution or they are correlated, a Monte Carlo simulation can provide a more reliable estimate of the final





uncertainty. However, a Monte Carlo approach is also computationally demanding with such an extensive dataset. We tested and compared both approaches for selected countries and sectors. Detailed information can be found in the SI, but the main conclusion is that we can mimic the results from the Monte Carlo simulation well with the uncertainty propagation functions. The methods show a similar order of magnitude and variability between countries and trace gases. Although there is no perfect

match between the two methods, we argue that this source of uncertainty is negligible compared to the uncertainty in the prior uncertainty data.

For the spatial proxies the same set of equations is applied, but to calculate the standard deviations weighted proxy maps are calculated. This means that for each combination of trace gas and country we determine the relative contribution of each sub-sector to the GNFR sector, assign a weight to the corresponding proxy map and multiply that value with the fraction in each

grid cell. This also results in a new weighted average proxy map per GNFR sector with a sum of 1 per country. Next, we calculate the uncertainty in this weighted average proxy map using Eq. (6), where the standard deviation is now related to the weighted fraction in each grid cell. The result of this is shown in Fig. 3.

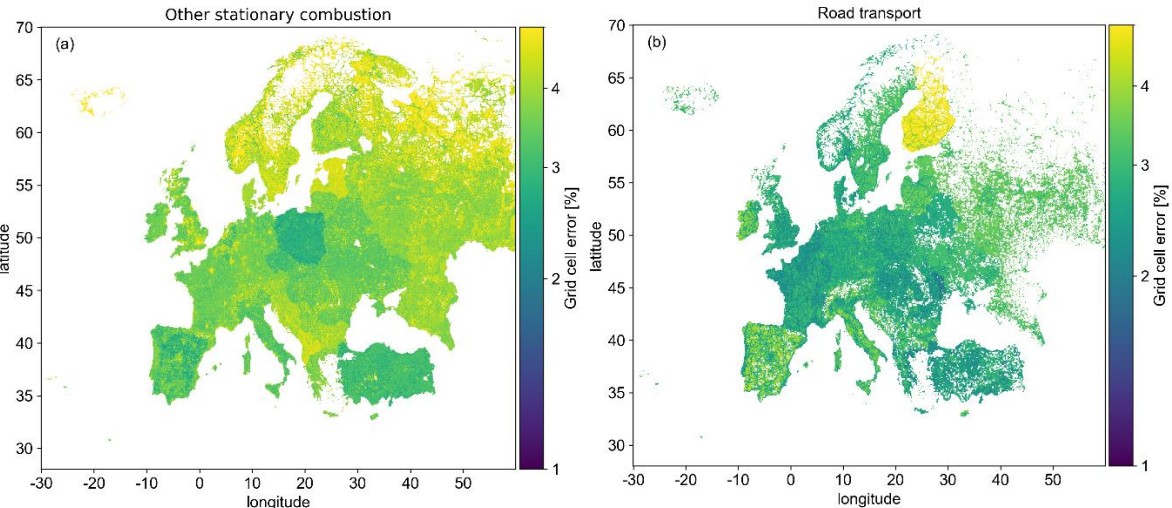

**Figure 3: Maps of gridded uncertainties [%] in CO$_2$ ff for other stationary combustion (a) and road transport (b).**

Finally, we determine the error correlation length for the GNFR sectors by calculating a weighted average correlation length. However, because the combined correlation length is also slightly sensitive to the uncertainty in each proxy map, the larger the uncertainty the more impact the spatial correlation has, we calculate the weight based on both the CO$_2$ emissions and the proxy map uncertainties (i.e., relative emission share multiplied by relative uncertainty share).

## 2.3 Inverse modelling approach

To examine the impact of the prior uncertainty definition on a multi-species inversion we perform a series of closed-loop numerical experiments. For this we generate a 'true' emission and use a chemical transport model to determine the atmospheric concentrations of CO$_2$ and CO that would be observed by the in-situ measurement network given these 'true' emissions (the





'true' observations). We perform an inversion where we confront a modelled atmosphere based on a prior estimate of emissions with 'real' observations ('true' observations with noise), adjusting the prior estimate to minimize the model-observation

differences. We can then compare the posterior and 'true' emissions to determine if our inversion approach is able to evaluate the accuracy of the prior estimate.

The analytical inversion approach is described elsewhere (e.g., Maasakkers et al., 2021), so we only briefly describe it here. We use the model to generate a Jacobian matrix ($\mathbf{K}$) that represents the observation sensitivity to emissions perturbations. We then use the minimization of the Bayesian cost function to solve for the posterior scale factors ($\mathbf{x}'$):

$$\mathbf{x}' = \mathbf{x}^a + \mathbf{S}^a\mathbf{K}^\mathrm{T}(\mathbf{K}\mathbf{S}^a\mathbf{K}^\mathrm{T} + \mathbf{R})^{-1}(\mathbf{y} - \mathbf{K}\mathbf{x}^a), \tag{7}$$

where $\mathbf{x}^a$ and $\mathbf{S}^a$ are the prior scale factors and error covariance matrix, respectively, and $\mathbf{y}$ and $\mathbf{R}$ are the observations and observing system error covariance matrix, respectively. In the following sections we describe the different aspects of the inversion system.

### 2.3.1 Atmospheric chemistry transport model

For the atmospheric chemistry transport model, we use GEOS-Chem version 12.5 (The International GEOS-Chem User Community, 2019). We model $CO_2$ and CO concentrations over Europe (15–35° E, 34–66° N) for the year 2018. The model is run at 0.25° x 0.3125° resolution and driven by GMAO GEOS-FP meteorology (Lucchesi, 2018). We use 3-hourly $CO_2$ and CO boundary conditions from a global simulation at 2° by 2.5° resolution. For anthropogenic $CO_2$ and CO emissions we use the TNO-GHGco-v4 inventory as described in Sect. 2.1.1. We use fire emissions from GFED v4 (Van der Werf et al., 2017),

biogenic fluxes from CASA-GFED (Ott, 2020), and ocean fluxes from Takahashi et al. (2009). For the inversion, we re-grid these emissions to basis functions (Fig. 4), which are created by aggregating regional emissions until a given emission threshold is reached, respecting country borders. We use pre-computed monthly, 3-D fields of the hydroxyl radical sink of CO. Further details on the model setup are provided elsewhere (Palmer et al., 2022; Scarpelli, in preperation). We assume a model uncertainty of 2 ppm and 8 ppb for $CO_2$ and CO, respectively.



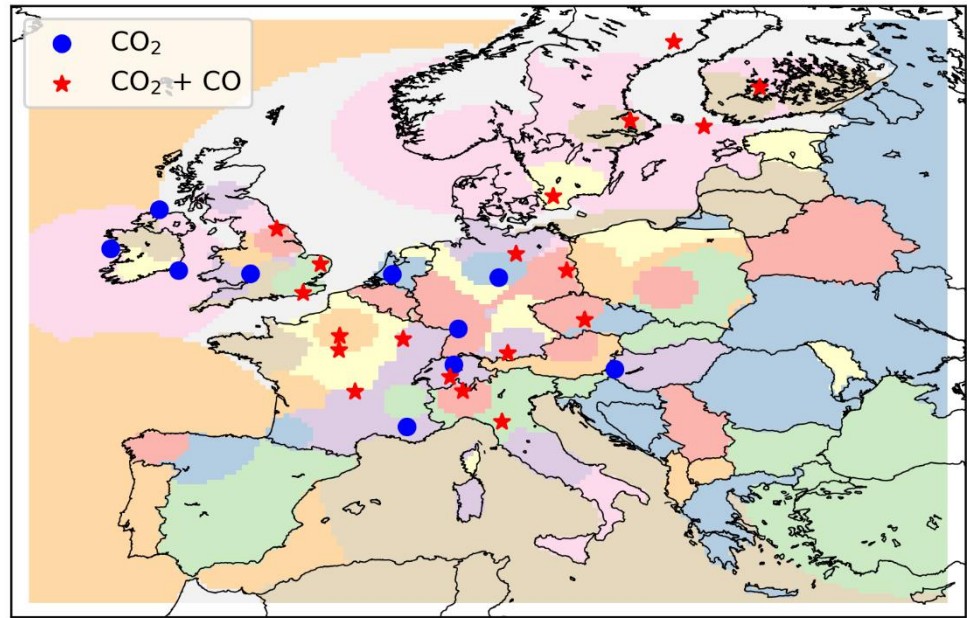


**Figure 4: Map of modelling domain, colours show basis functions. The blue dots represent locations where CO₂ is measured, the red stars represent locations where both CO₂ and CO is measured.**

### 2.3.2 State vector and error covariance matrix

The state vector consists of scale factors for the fossil fuel ($\mathbf{x}_{co2}{}^{FF}, \mathbf{x}_{co}{}^{FF}$) and biogenic ($\mathbf{x}_{co2}{}^{Bio}$) components, boundary

conditions ($\mathbf{x}_{co2}{}^{BC}, \mathbf{x}_{co}{}^{BC}$), and CO chemistry ($\mathbf{x}_{co}{}^{Chem}$) terms:

$$\mathbf{x} = \left(\mathbf{x}_{co2}{}^{BC}, \mathbf{x}_{co}{}^{BC}, \mathbf{x}_{co}{}^{Chem}, \mathbf{x}_{co2}{}^{Bio}, \mathbf{x}_{co2}{}^{FF}, \mathbf{x}_{co}{}^{FF}\right). \tag{8}$$

These scale factors are optimized per basis function (Fig. 4) and per month. We assume a prior Gaussian uncertainty of 50 %,

5 %, and 5 % for the biogenic, boundary condition, and CO chemistry scale factors, respectively.

For the fossil fuel state vector elements ($\mathbf{x}_{co2}{}^{FF}, \mathbf{x}_{co}{}^{FF}$), we use a Monte Carlo approach to determine the prior uncertainties,

taking advantage of the advanced uncertainty estimate presented here. Separate ensembles are made for the spatial distribution

and the country-level emissions, which are combined into one ensemble of gridded emissions and fed into the inversion system.

First, we generate an error covariance matrix of country-level emissions, where each element corresponds to a single GNFR

sector and species (CO₂ or CO). We use the standard deviations derived in Sect. 2.2.3 ($\sigma_x$) to populate the diagonal of the

covariance matrix, whereas all off-diagonal values are zero (no error correlations).

Second, for a given GNFR sector, we generate an error covariance matrix for the spatial distribution using the uncertainties

for the proxy maps described above. Each sector's error covariance matrix includes both CO and CO₂. The variances on the

diagonal of the matrix are derived from the standard deviations described in Sect. 2.2.3 (and shown in Fig. 3). The off-diagonals

of the error covariance matrix include the covariance between spatially neighbouring grid cells that belong to the same species



(CO or $CO_2$), derived from the spatial error correlations described in Sect. 2.2.1, and the covariance between CO and $CO_2$

gridded emissions, derived from Sect. 2.2.2. For the error covariances within a single species we define the covariances based on the spatial error correlation length $l$. For this we define an exponential decay in the correlation coefficient $r$ between elements $i$ and $j$ with distance $d$ (Eq. (9)). After distance $l$ we assume the correlation is zero, following Kunik et al. (2019).

$$r_{i,j} = e^{-d_{i,j}/l}. \tag{9}$$

We perform a Cholesky decomposition of each error covariance matrix, resulting in a matrix $\mathbf{L}$. Combining this matrix with a

vector of uncorrelated random samples ($\boldsymbol{u}$) from a Gaussian distribution with $\mu = 0$ and $\sigma = 1$ through a dot product gives us a perturbation vector with the covariance properties of the whole system ($\boldsymbol{p}$). We can do this for $m$ unique perturbation vectors to generate an ensemble of $m$ spatial distributions or country-level emissions ($\boldsymbol{x_m}$):

$$\boldsymbol{p_m} = \boldsymbol{L} \cdot \boldsymbol{u_m}, \tag{10}$$

$$x_m = \bar{x}p_m + \bar{x}, \tag{11}$$

where $x_m$ is the estimated values of the spatial map for a given sector (includes $CO_2$ and CO) for ensemble member $m$, and $\bar{\boldsymbol{x}}$ is the expected value of the spatial distribution.

Or for variables with a lognormal distribution, we calculate the ensemble values with:

$$x_m = \bar{x}e^{p_m}. \tag{12}$$

The ensemble of gridded emissions is a combination of the ensemble of spatial distributions and country-level emissions.

### 2.3.3 Observations

We generate 'true' emissions by perturbing the prior emission inventories based on assumed error statistics, as described previously, assuming that the previously described error correlation between $CO_2$ and CO is true. For the 'true' observations, we sample the 3-D modelled concentration fields as observed by the in-situ network (Fig. 4) and because our system is linear we can apply the same perturbations to the observation vectors as we applied to the 'true' emissions. The 'true' observations

are the $CO_2$ and CO concentrations that would result if the 'true' emissions occurred. We generate our 'real' observations by adding a noise term to the 'true' observations, simulating what the observing network would have generated had the 'true' emissions occurred. The noise term is a vector of perturbations taken from a Gaussian distribution of mean one with a standard deviation of 2 ppm and 4 ppb for $CO_2$ and CO, respectively, and represents the observation uncertainty.

### 2.3.4 Experiments

To illustrate the impact of the new prior uncertainty definition we also report the results from a second inversion approach for which we assume there is 100 % error correlation between $CO_2$ and CO emissions from fossil fuel combustion, allowing the use of one shared fossil fuel scale factor for both $CO_2$ and CO ($\mathbf{X}^{FF}$). This uncertainty definition has been used before by Palmer et al. (2022) and serves as a base experiment. For this test scenario, we use a combustion uncertainty of 10 % for the





whole domain and a spatial error correlation length of 20 km. For comparison, the mean prior uncertainty in the advanced

experiment is 7.7 % for $CO_2$ and 11.8 % for CO. Finally, we do the same numerical experiments without CO.

**Table 3: Overview of inversion experiments. Advanced uncertainties and CO:CO2 error correlations are those developed here. Simple uncertainties refer to a fixed 10 % combustion uncertainty. These are combined with a full CO:$CO_2$ error correlation, i.e., one scaling factor applies to both $CO_2$ and CO fossil fuel fluxes.**

| Experiment name | Included trace gases | Uncertainties | CO:CO2 error correlation |
|---|---|---|---|
| **Adv_CO2_CO** | $CO_2$ and CO | Advanced | Advanced |
| **Adv_CO2** | $CO_2$ | Advanced | n/a |
| **Base_CO2_CO** | $CO_2$ and CO | Simple | 100 % |
| **Base_CO2** | $CO_2$ | Simple | n/a |

## 3. Results

### 3.1 Assessment of prior uncertainties and error correlations

First, we show the results from the prior uncertainty calculations, before reporting results from the closed-loop numerical experiments. The spatial error correlation lengths calculated per proxy map per country are shown in Fig. 5, including the median value for all countries. The resulting correlation lengths are also given in Table 2.



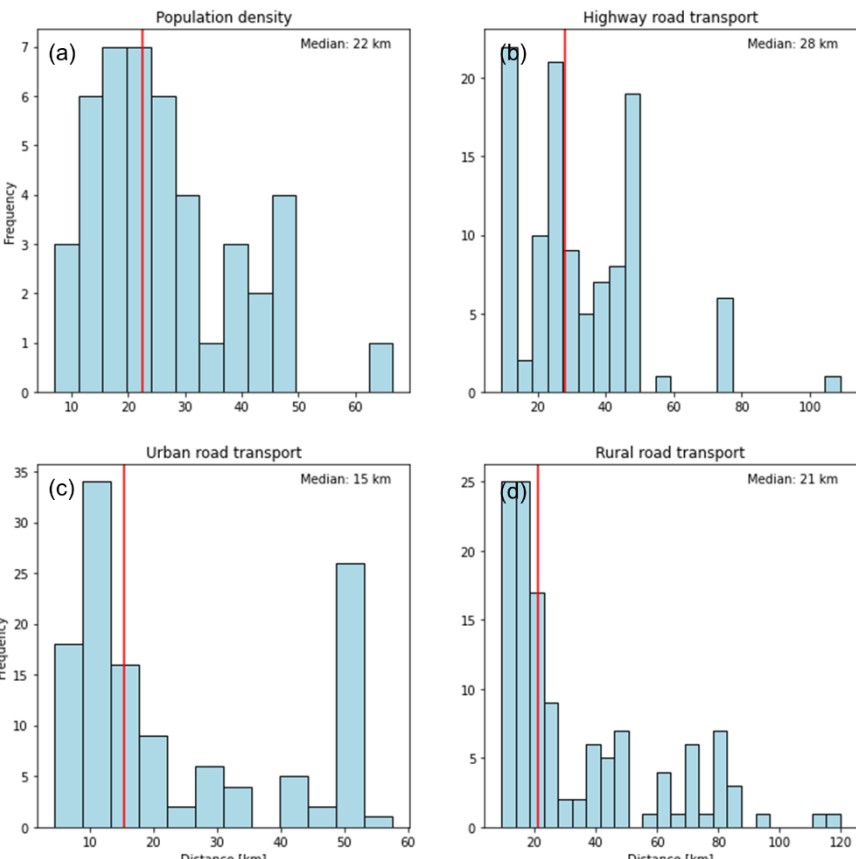

**Figure 5: Derived correlation lengths for the European proxy map of (a) total population density and road transport proxy maps per road type (b–d) for all vehicle types combined, binned per 5 km. The red line shows the median value.**

For population density there are large differences between country groups, for example between North/South and East/West Europe. The clustering of people in cities and the rural space within those broad geographical regions differs on a regional basis and affects the correlation lengths accordingly. Nevertheless, the largest group of countries shows correlation lengths of less than 30 km, which given the resolution of the data assimilation system is relatively small (~5 pixels). For wood use, which we use as a proxy map for residential biomass combustion, we see a large cluster around 20–30 km (not shown) and there are only few countries with significantly different length scales. For the road transport proxy maps the various vehicle types (passenger cars, light-duty vehicles, heavy-duty vehicles) do not show much variability in correlation lengths, but differences are evident for different road types; consequently, we combine the vehicle types to obtain road transport correlation lengths per road type. This results in longer length scales for highways than for urban roads. In urban areas short distances are covered more frequently, resulting in weaker correlations in road transport activity between locations.

Next, we predict the CO:CO$_2$ error correlation that results from the shared activity between the trace gases. The relationship between the Monte-Carlo based CO:CO$_2$ error correlations and the predictor (Eq. 3) is shown in Fig. 6. As mentioned before,



the Monte-Carlo simulation is performed for seven selected countries. We find a clear cosine-shaped relation for GNFR C, so

that the correlation coefficient can be estimated with the equation:

$$r = a \cdot \cos(b \cdot P_{GNFRC}),$$   (13)

where $a$ and $b$ are parameters estimated from the fit shown in Fig. 6. The $a$ parameter denotes the highest possible correlation

coefficient, which for pixels with emissions from only one sub-sector should be (close to) 1. For the seven individual countries

the $a$ parameter lies between 0.96 and 1.03. The $b$ parameter is the period of the cosine function, which indicates how sharply

the function declines with increasing predictor values. This parameter is between 3.36 and 4.44 for these seven countries. The

mean for all countries is 3.60.

We see some grid cells with a predictor value of zero, whereas the correlation coefficient is much lower than the $a$ parameter.

In these cases, there are two proxy maps with similar shares for CO and $CO_2$ (hence, a very small standard deviation). These

cases mostly occur in Sweden, where it results in a relatively poor fit of the cosine function ($R^2$ of 0.46). Overall, these cases

make up 0.1 % of all grid cells and they have no significant impact on the definition of the average function, which has an $R^2$

of 0.85. The fit of the other individual countries is between 0.79 and 0.98. Plots for individual countries are shown in the SI

(Fig. S3).

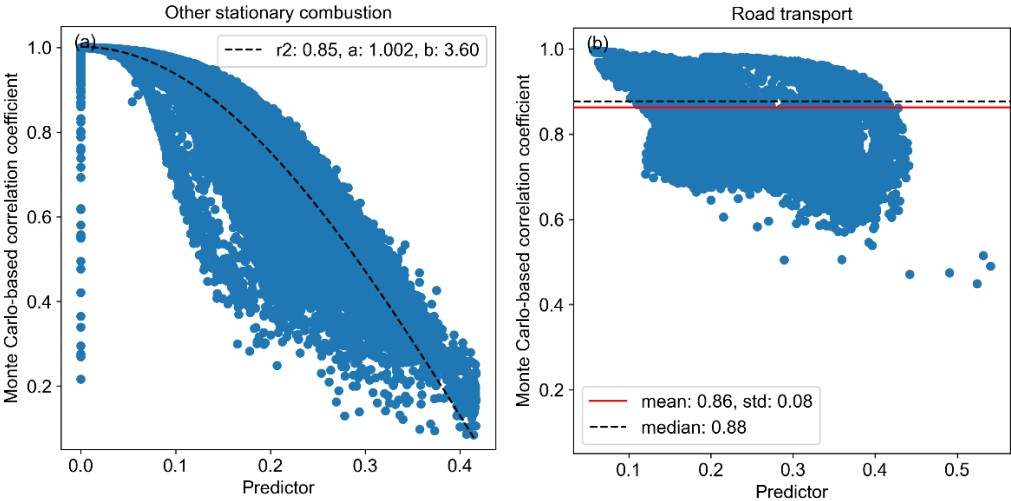

**Figure 6: Scatter plots of Monte-Carlo (N=500) based correlation coefficient (r) per grid cell against the predictor calculated with**
**Eq. (3). In panel (a) the fit ($R^2$) and cosine function parameters are shown. In panel (b) the mean, median and standard deviation**
**(std) of the correlation coefficients are shown.**

For road transport it is more difficult to extract a relationship between the predictor and the correlation coefficient. For

individual countries we mostly see several cosine-shaped structures (shown in the SI, Fig. S4), which makes it impossible to

identify one single function. To understand this behaviour, we looked in more detail at the vehicle and road types. Although

different vehicle types show very similar cosine functions within a country (Fig. 7), when we combine them the structure

disappears. The relationship between the predictor and the correlation coefficient seems to depend not only on the vehicle type,

but also on the amount of road types that are present in a grid cell. When combining all road and vehicle types they start to



affect each other, so within the scatter plots we can no longer identify vehicle types. Because we are not separating vehicle and road types in our prior uncertainty data for the data assimilation system, we only want one value. Compared to the other

stationary combustion sector we see much less variability in the correlations and therefore we use the median value of 0.88. For individual countries the median value lies between 0.82 and 0.96.

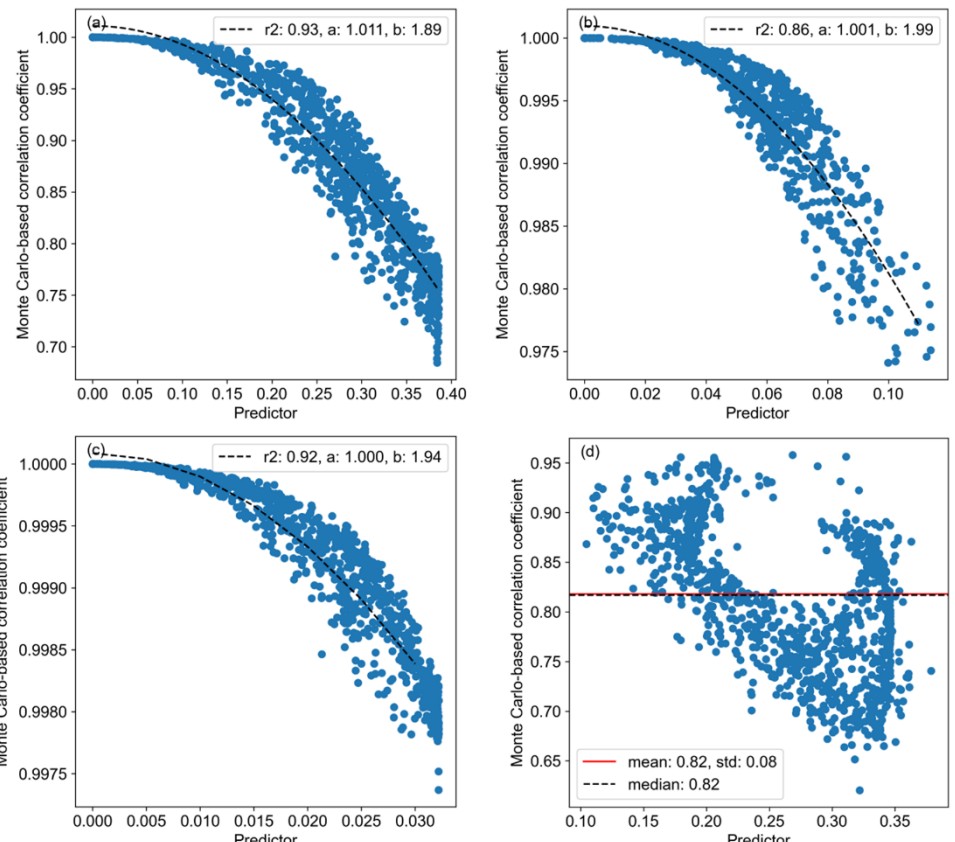

**Figure 7: Scatter plots of Monte-Carlo (N=500) based correlation coefficient (r) per grid cell against the predictor calculated with Eq. (3) for the Netherlands for GNFR F per vehicle type (PC: passenger cars; LDV: light-duty vehicles; HDV: heavy-duty vehicles).**
**The correlation ($R^2$) and cosine function parameters per vehicle type are also shown. The bottom right panel shows the same scatter plot when all vehicle types are combined, including the mean, median and standard deviation (std) of the correlation coefficients.**

### 3.2 The effect of the prior uncertainty definition

Next, we examine the impact of including this advanced definition of gridded emission uncertainties and error covariances on our ability to estimate combustion $CO_2$ emissions in an inversion framework. Figure 8 shows the annual average difference

between the absolute prior and posterior deviations from the true emissions. Positive (negative) values, represented by red (blue) colours, indicate that the posterior is closer to (further from) the truth than the prior. With the base uncertainties there are several areas in which the results deteriorate (blue colours), such as the Netherlands, the south-east of the UK and some locations in Germany. The differences range -4.59–10.87 kg s$^{-1}$ and -5.13–11.99 kg s$^{-1}$ for the Base_CO2 and Base_CO2_CO



experiments (Table 3), respectively. When using the advanced uncertainties these blue colours start to disappear and the differences range -2.58–12.31 kg s$^{-1}$ and -1.60–12.92 kg s$^{-1}$ for the Adv_CO2 and Adv_CO2_CO experiments, respectively. The average $CO_2$ fossil fuel flux in this domain is 7.55 kg s$^{-1}$, with a maximum value of just over 2000 kg$^{-1}$. Hence, the relative differences are small, but nevertheless show a consistent improvement with the advanced uncertainties.



**Figure 8: Map of prior - posterior annual average absolute deviation from the truth in fossil fuel $CO_2$ emissions for all four experiments. Unit is kg s$^{-1}$ per grid cell. Note that the bounds of the colour bars are set to -2.5–2.5 kg s$^{-1}$.**

Generally, there seem to be fewer areas with significant deviations from the prior when using the advanced uncertainties, and regions that still show differences mostly show an improvement. This suggests that with the advanced uncertainties the system has greater ability to constrain fossil fuel $CO_2$ emissions. This is also illustrated by the reduced posterior error correlation between the $CO_2$ biogenic and fossil fuel fluxes (Fig. 9), which is also seen for individual months. Note that the prior error correlations between biogenic and fossil fuel $CO_2$ fluxes are zero. A high posterior error correlation means that the inversion





system is unable to assign model-data mismatches to specific sources, but rather updates multiple scaling factors at once, which has a lower cost.

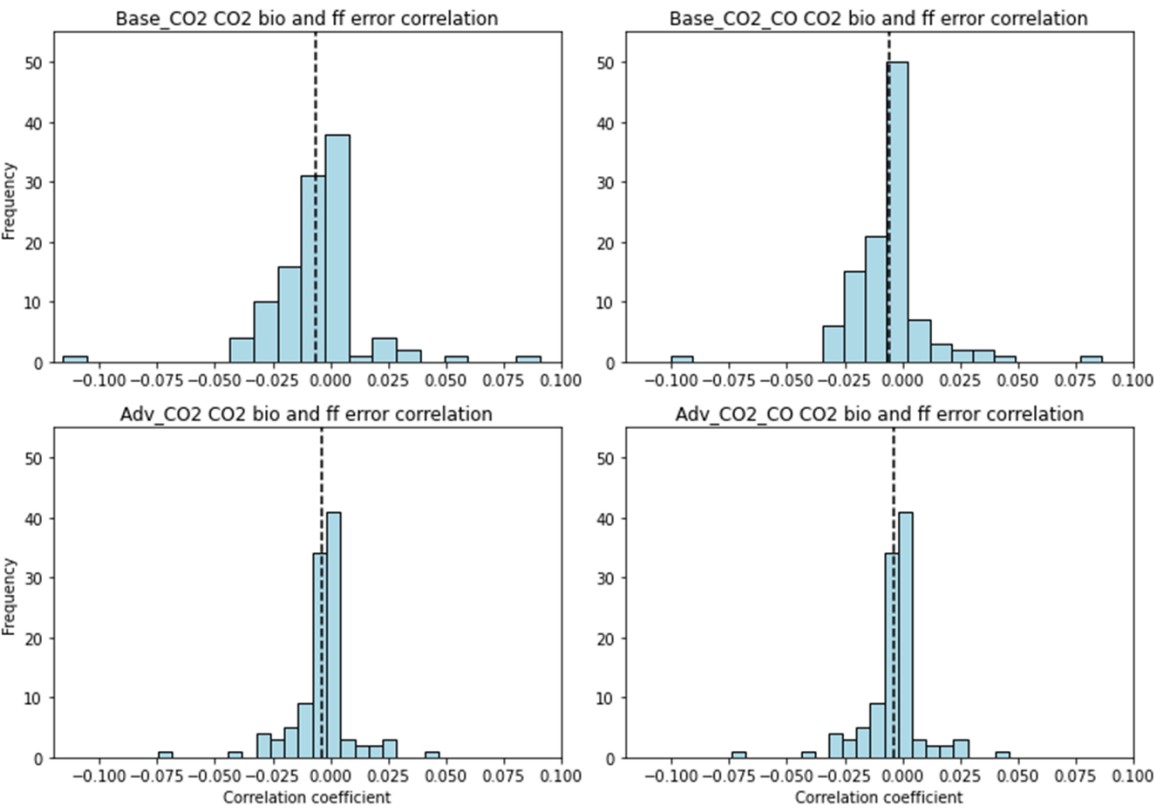

**Figure 9: Histograms showing posterior error correlations between CO₂ bio and CO₂ ff for all basis functions for three experiments. Mean values are shown with a dashed vertical line.**

### 3.3 Combining CO and CO₂

Figure 8 also shows that adding CO in the experiment with the base uncertainties causes more areas to show significant differences between the prior and posterior. Moreover, on average the deviations from the prior are larger as well. Since the main source of CO is fossil fuel combustion and the uncertainties in CO emissions are large, this pollutant is more sensitive to errors in prior emissions and therefore causes larger deviations from the prior. This indicates that CO adds additional information to the system on fossil fuel fluxes, which causes the model-data mismatch in $CO_2$ to be assigned more clearly to either the $CO_2$ biogenic or fossil fuel fluxes. This is also illustrated by Fig. 9, which shows the posterior error correlations between $CO_2$ biogenic and fossil fuel fluxes. When adding CO to the base experiment the mean correlation per basis function is closer to zero. This also results in a slight improvement in the posterior scaling factors for $CO_2$ biogenic fluxes (Fig. 10).





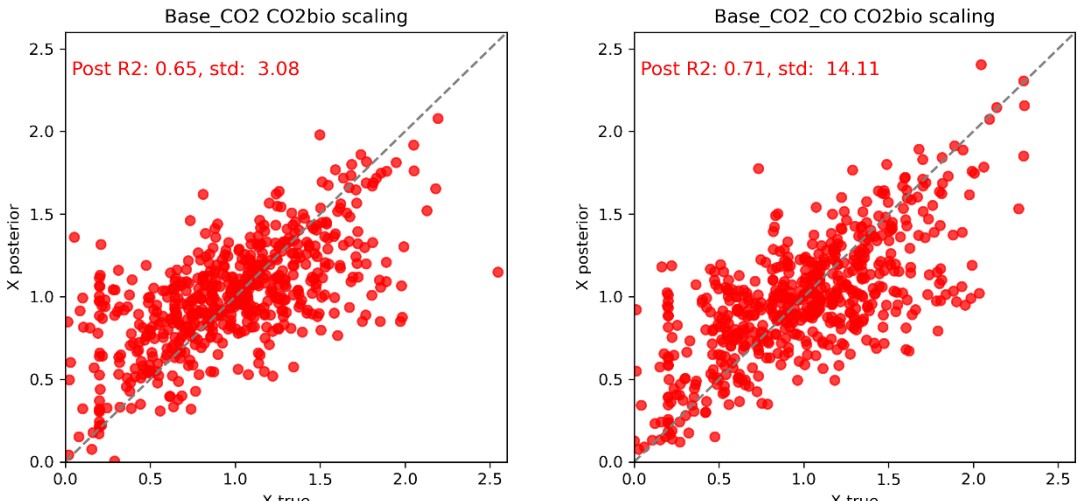

**Figure 10: Scatter plot of true vs. posterior scaling factors for $CO_2$ bio. The correlation coefficient ($R^2$) and standard deviation are also given.**

However, Figure 8 shows more blue colours for Base_CO2_CO than for Base_CO2, for example in northern Italy, which

means that the results for $CO_2$ fossil fuel fluxes are actually worse than when using only $CO_2$. This pattern is not visible when comparing the Adv_CO2 and Adv_CO2_CO experiments. In other words, adding CO can deteriorate the results for experiments in which the prior error correlation is not correctly defined due to the sensitivity of CO to assumed prior emission uncertainties. Using a full CO:$CO_2$ error correlation causes larger changes in the scaling of $CO_2$ fossil fuel fluxes, because the uncertainties in CO are relatively large and can be scaled easily. $CO_2$ then follows suit, which is clearly not always correct.

With the advanced uncertainties there are fewer large changes when comparing the experiments with and without CO. But there are some small improvements visible in the UK and results show no spurious changes in $CO_2$. Hence, in the combined optimization of $CO_2$ and CO there is a clear need for advanced uncertainties to prevent inaccurate emissions corrections.

## 4. Discussion and conclusions

We presented here a detailed assessment of prior emission uncertainties to support data assimilation studies. The prior

uncertainties have a significant impact on data assimilation, as they determine the extent to which the prior emissions can be corrected. Underestimating the uncertainties limits the freedom of the system to correct the prior, which means that the actual state can be outside the uncertainty range and therefore be unreachable. Overestimating the uncertainties reduces the constraint from the prior information, meaning that we do not make optimal use of the prior knowledge given to the system. Moreover, a prior uncertainty definition that includes covariances enables us to use co-emitted species to estimate fossil fuel $CO_2$. Hence,

a realistic prior uncertainty definition is important.



Building on the work of Super et al. (2020a) we developed a prior uncertainty definition that is based on the uncertainties in the underlying data used to create the emission inventory. This ensures that the uncertainty definition is fully consistent with the emissions and consistent across multiple species (here, $CO_2$ and CO). We presented a more detailed analysis of the spatial uncertainties, including a description of spatial error correlation lengths. We particularly focused on the $CO_2$:CO error

correlations, which are caused by the shared activity resulting in emissions and mainly manifests itself in the spatial patterns. An important source of uncertainty in this work is the detailed uncertainty data that we use as a starting point, i.e., the reported emission uncertainties and the uncertainties in the spatial proxies. For GHGs the reported country-level uncertainties are used. The IPCC encourages countries to make country-specific uncertainty assessments based on expert judgement, but also provides default options (Eggleston et al., 2006). Therefore, the reported uncertainties are not necessarily consistent between countries.

Nevertheless, we adopt these reported uncertainties to ensure our uncertainty estimates are well-documented and consistent in methodology. For the spatial proxies the uncertainties are also based partly on expert judgement in the absence of a better quantification. For the representativeness error we assume a similar order of magnitude as the proxy value uncertainty, which is an arbitrary choice. Hogue et al. (2016) estimated the uncertainty of using population density as a proxy for $CO_2$ emissions by comparing differences in the emissions per capita for all US states. They find that this is often the dominant source of

uncertainty and hence we argue that our estimate is conservative. Since we start at a high level of detail, we assume some fraction of the random errors in the prior uncertainty information will cancel out. Moreover, we ignored the proxy quality as a source of uncertainty. We evaluate our approach by comparing the results against previous work. The country-level and grid cell uncertainties differ only slightly from the results of Super et al. (2020a). These results are discussed in detail there and here we only focus on the spatial error correlation lengths and $CO_2$:CO error correlations. We evaluate the overall prior

uncertainty definition using closed-loop numerical experiments, which we discuss below.

The spatial error correlation lengths have been estimated by fitting semi-variograms to the proxy data and range between 15 and 28 km, which is about 2.5–4.5 times the grid size. Kunik et al. (2019) used a similar approach to estimate the length at which the difference between two emission inventories was still correlated. They found a correlation length scale of 6 km, which is about 6 times the grid size. Other studies optimized the correlation length based on the spatial scale and resolution of

their inversion and the density of the observation network. Generally, a larger spatial correlation length means a larger aggregated uncertainty and therefore a larger correction to the observations is possible. Hence, this length scale can be optimized statistically. For example, Lauvaux et al. (2016) examined the impact of the spatial correlation length on inversions to estimate $CO_2$ fluxes from the city of Indianapolis in the US. They found that ignoring the spatial correlation resulted only in local emission adjustments around the measurement sites, because areas further from those sites are not constrained by the

observations. Increasing the correlation length to 12 km adjusts the emissions for the whole city at once and the spatial patterns are not affected. They concluded that a correlation length of 4–5 km is most suitable to make optimal use of the observations and prior information (Nathan et al., 2018). Similar conclusions were drawn for $N_2O$ on a European scale (Corazza et al., 2011) and for biogenic $CO_2$ fluxes (Lauvaux et al., 2012). Of course, the optimal length scale based on this approach depends strongly on the spatial scale considered and we consider our correlation lengths to be relatively low compared to the observation



network. Based on these findings we argue that it is necessary to combine the data-driven estimate with a statistical approach to find an optimal correlation length. Unfortunately, a methodology for this is not yet existent. Moreover, the spatial correlation length scale may depend on the considered time scales (Carouge et al., 2010). Hence, more work is needed on this topic.

We developed a new approach to define the $CO:CO_2$ error correlation. Previous studies have often assumed a perfect correlation between the errors in $CO_2$ and CO fossil fuel fluxes, for example through a fixed emission ratio (Brioude et al.,

2013; Nathan et al., 2018). However, emission ratios have a large uncertainty and therefore the $CO_2$ and CO errors are not perfectly correlated. Only few studies have tried to estimate the inter-species error correlation or performed sensitivity tests. Palmer et al. (2006) tried to make use of the synergy between CO and $CO_2$ by calculating error correlations per country. These correlations are very small, because the CO EF dominated the uncertainty but is uncorrelated with $CO_2$. They concluded that the error correlation should be larger than 0.5 for CO to be a useful constraint for $CO_2$ fluxes, which is unrealistic at the

country-level. However, for gridded emissions the correlation is much stronger as spatial patterns are linked to the activity. Moreover, the correlation is larger for individual sectors. Therefore, we argue that our calculated grid cell correlations between 0.18 and 0.99 (with a mean value of 0.89) are realistic, considering they are sector-specific and gridded. Additionally, Boschetti et al. (2018) tested different correlation strengths $(0.1 - 0.9)$ and found no significant difference in the posterior fluxes, although the uncertainty reduction increased with stronger correlations. This makes sense because it means more information is taken

from CO priors and observations. We also find larger uncertainty reductions when we add CO to the base case, i.e., with a $CO:CO_2$ error correlation of 1. However, we also illustrated that the results do not always improve. The reason could be that Boschetti et al. (2018) assume one error correlation value for the whole domain and also evaluate their results for the whole domain. Given the ranges in the prior – posterior absolute uncertainties from Fig. 8 for the Base_CO2 and Base_CO2_CO experiments we also see no significant difference in the domain total emissions. However, we see clear differences per region.

Closed-loop numerical experiments are useful for evaluating the capability of observing systems, including assumed prior and measurement error covariance matrices, to determine accurate estimates of carbon fluxes (Masutani et al., 2010). However, they also have limitations. The theoretical impact of an observing system will depend on several factors, including the quality of the atmospheric transport model used, the assumed structure and values used by the assimilation error covariance matrices, and the spatial distribution of the observations. Some of these choices can be based on expert judgement. For our numerical

experiments, we are also limited to the resolution of our basis functions and it is likely that we would see greater benefit from the Adv_CO2_CO experiment if the inversion were performed at high resolution, leveraging the fine-scale variability in the error correlations between CO and $CO_2$ (e.g., along road networks).

Our numerical experiments illustrate the impact of the prior emission uncertainties. From them, we can draw two important conclusions: 1) the prior uncertainty definition is important to differentiate between different fluxes, such as biogenic and

fossil fuel $CO_2$, and 2) CO can provide an additional constraint to estimate fossil fuel $CO_2$ fluxes only if the error covariance structure is defined realistically. Generally, it is difficult to constrain $CO_2$ fossil fuel flux estimates due to the high uncertainty in biogenic fluxes. However, we show here that with the improved uncertainty definition the posterior error correlation between biogenic and fossil fuel $CO_2$ is weaker. A likely explanation for this is that the largest fossil fuel sources are often clustered in



different areas than those with the largest biogenic fluxes. Hence, when describing the spatial error structure correctly the
estimation framework used within the numerical experiments can more easily detect which source is dominant and update the
estimates accordingly. Additionally, we have shown that CO adds additional information on the $CO_2$ fossil fuel fluxes in the
base experiments, whereas it has a minor impact on the experiments with the updated prior uncertainties. Since CO has
relatively large prior uncertainties (in emissions, model and observations) compared to $CO_2$, the prior and observational
information of CO contributes little weight to the cost function. By setting the $CO:CO_2$ error correlation to 1, the CO
information becomes more important and thus results in larger corrections. The $CO:CO_2$ error correlations in the advanced
experiment are relatively high (e.g. 0.88 for road transport), so this is likely not the only reason. It is likely that a better
definition of the prior uncertainties helps to better weigh all the information and therefore address some of the spurious changes
seen in the Base_CO2_CO experiment.

In this study we have used in-situ observations across the UK and mainland Europe, which have a limited spatial coverage
with 29 stations measuring $CO_2$, of which 19 also measure CO. Figure 8 shows that fluxes are mostly altered in central Europe,
where the observation network is densest and therefore enough information is available to update the prior emission data. This
stresses the need for a wide observation network, ideally with co-located observations of $CO_2$ and co-emitted species. For this
reason, we also argue that the added value of CO is likely more pronounced with satellite data. The $CO_2$ column observed by
satellites has limited sensitivity to $CO_2$ emissions perturbations, so our ability to constrain fossil fuel $CO_2$ fluxes separately
from biogenic fluxes is limited. For co-emitted species CO and $NO_2$, the atmospheric column has higher sensitivity to
perturbations of combustion emissions, adding value to their inclusion in the inversion (Konovalov et al., 2016; Liu et al.,
2020; Nathan et al., 2018; Reuter et al., 2019). In addition, satellite instruments like TROPOMI provide high density
observations of CO and $NO_2$ globally, increasing the information content of the inversion compared to a $CO_2$-only inversion,
whereas for the in-situ network used here we have fewer in-situ stations with CO observations compared to $CO_2$.

Finally, our work illustrates the importance of using an accurate definition for prior uncertainties for the $CO_2$ inversion and
the multi-species inversion. A concerted effort is needed to quantify the prior uncertainties in a way that is consistent with the
data and optimized for its application in data assimilation studies. There is much room for further improvements of current
work, for example, by adding more detailed uncertainty estimates for other sectors than road transport and other stationary
combustion. Given the nature of the spatial proxy maps of the other sectors, which are often categorial or with point source
locations, this poses an additional challenge. Furthermore, temporal uncertainties need to be added, as they can have a major
impact on data assimilation results (Super et al., 2021).

**Code availability**

The codes used for analysing and plotting the results from the numerical experiments are accessible through Zenodo (Super et
al., 2024).



**Data availability**

Most of the data used to quantify emission uncertainties comes from public sources. The CAMS-REG emission inventory is accessible via https://permalink.aeris-data.fr/CAMS-REG-ANT (Kuenen et al., 2021). Access is provided through the Emissions of atmospheric Compounds of Ancillary Data (ECCAD) system. The advanced uncertainties prepared as part of this manuscript are accessible through Zenodo (Super et al., 2024).

**Author contribution**

IS and AD gathered and processed all the prior uncertainty data and developed the methodologies for quantifying error correlations. Refinements were made in consultation with TS and PIP. The closed-loop numerical experiments were set up and performed by TS, in consultation with PIP, IS and AD. The experiments were analysed by IS and TS, and discussed with AD and PIP. IS prepared the paper, with specific input sections from AD and IS. PIP, TS and AD reviewed the paper as a whole
prior to submission.

**Competing interests**

The authors declare that they have no conflict of interest.

**Acknowledgements**

The research leading to these results has received funding from the CoCO2 project. The CoCO2 project has received funding
from the European Union's Horizon 2020 research and innovation programme under grant agreement No 958927

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
