# Peer review of "Improved definition of prior uncertainties in CO2 and CO fossil fuel fluxes and the impact on a multi-species inversion with GEOS-Chem (v12.5)"

_EGUsphere, 2023_

## Author Response (AR1)

We would like to thank the reviewers for their enthusiasm about our study and for the comments on our work. The review comments have been helpful in reflecting on our work and pointing out parts that required further improvements. Below we address specific issues mentioned by the reviewers point by point. The manuscript has been updated accordingly (changes are highlighted, line numbers refer to the track-changes manuscript).

***Fabian Maier, 21 Apr 2024***

*This paper gives a detailed estimation of prior uncertainties in CO2 and CO fossil fuel fluxes. This includes the derivation of spatial error correlation lengths and CO2:CO error correlations from an in-depth analysis of spatial proxy maps. Synthetic CO2-CO multi-species inversion experiments reveal the importance of advanced prior uncertainty definitions to more reliably differentiate between biosphere and fossil fuel CO2 fluxes and to gain advantage from CO in constraining fossil emissions.*

*Overall, this is a very relevant and good study, which provides the basis for exploiting similarities in the CO2 and CO emission patterns with multi-species inversions. The manuscript is well and illustratively written. Please find below some suggestions for clarification. I recommend acceptance with these minor revisions.*

*Minor comments:*

*Estimating and compiling all the uncertainties shown in this study is a tough task. The authors have done this very carefully. However, as also mentioned in this study, some uncertainty estimates are based on expert judgements; e.g., it was assumed that the representativeness error is of a similar order of magnitude as the proxy value uncertainty. I wonder how this choice affects the results of the closed-loop experiments, considering that the representativeness error may cause spatial correlations. Maybe this could be addressed by performing a sensitivity study with representativeness errors that are slightly varied. However, I do not know how easily such an analysis can be conducted.*

We have performed an experiment in which we excluded the representativeness error (i.e., the spatial errors are divided by two). In that case we still see the same differences between experiments: the advanced experiments show more areas with improvements and adding CO mainly reduces the performance with the base uncertainties. However, the corrections from the truth are about ten times smaller for all experiments. This illustrates that the spatial errors play a very important role and assumptions we make in that regard have a significant impact on the results. This was our motivation to make a conservative estimate of the spatial errors to not overestimate the impact of our work, as also mentioned in the discussion (lines 559-560). However, we do believe that our conclusions on the impact of adding CO and having a better error definition hold, also when the errors are a bit too low or too high.

*In Sect. 3.3 the authors nicely describe and explain the results of the closed-loop experiments, which are also shown in Fig. 8 – 10. However, I think this section would benefit from some numbers or statistics to back up the statements made. I would also be interested to know whether the differences between the CO2 only and CO2 & CO experiments are significant.*

Significant differences in the maps in Fig. 8 are difficult to establish. When looking at the basis functions we see regions where the experiments with CO show improvements, whereas the experiments without CO show deteriorations, or vice versa. Other regions show corrections in the same direction for experiments with and without CO. So drawing conclusions on significance based on these data depends strongly on which variables and regions you look at. We can generally say that we see certain signals, but given the constraints of the measurements (as discussed in the manuscript) we cannot draw statistically significant conclusions from our work. We did however add some information on the number of grid cells with differences between the prior and posterior error of more than 1 kg s$^{-1}$ and the share showing worse results in the posterior compared to the prior (lines 496-498 and 527-530). We have also added a sentence on the relative errors in the prior/posterior for the different experiments (lines 487-489). We hope to have fulfilled the reviewer's request with this.

*Unfortunately, I could not fully follow the calculations of the predictor P in Sect. 2.2.2. I think this section needs some revision. Most importantly, please explain for what the vector W stands and indicate over which elements the sums in the denominators in Eq. 2 are calculated. Please also see my specific questions/remarks below. More generally, the predictor was introduced to estimate the error correlation between CO and CO2. Thus, I would*

*have expected a positive correlation between predictor value and the Monte-Carlo based correlation coefficients. However, it turns out that the Monte-Carlo based correlation coefficients decrease with increasing predictor values (e.g. Fig. 6a). What is the reason for this? Explaining this in more detail might also help the reader to better understand the predictor definition.*

The predictor is a measure of dissimilarity between $CO_2$ and CO and therefore the correlation coefficient decreases with increasing predictor values. It is simply a matter of how the predictor was defined. We have added the definition to the manuscript to guide the reader (lines 262-263). We have addressed the reviewer's specific concerns below in order to increase understanding of how the predictor was calculated.

*Specific comments:*

*Abstract:*

*L. 26-30: Can you strengthen these statements by providing some numbers? In order to interpret and classify your results, it may be useful to briefly describe here how the true and prior values were determined for your synthetic experiment.*

We have added information on the number of grid cells with differences between the prior and posterior error of more than 1 kg s$^{-1}$ and the share showing improved results in the posterior compared to the prior (lines 29-34). We also added a short description of the prior and true emissions (lines 24-26).

*Introduction:*

*L. 51ff: If you like, you could also mention 14C here, which is the gold standard for separating fossil fuel CO2. You could then further motivate the usage of co-emitted species like CO and NOx by referring to the typically low temporal resolution and poor spatial coverage of the 14C measurements.*

We thank the reviewer for this suggestion and have added a sentence on radiocarbon (lines 62-64).

*L. 73: Since the term "emission factor" is used frequently in your study, it may be useful to provide a brief definition and/or its units here.*

An explanation of the term "emission factor" is added (lines 78-79). However, we cannot provide units, as these can differ depending on how the activity is defined.

*L. 77-78: The last part of this sentence is a bit confusing. Do you mean "the emission factor of those (emission) sectors for which the error is not correlated to the error in the CO2 emission factor"?*

We have updated this sentence:

"In order to simultaneously optimize $CO_2$ and CO emissions we need to make optimal use of these synergies. At the national scale the most uncertain parameter is the CO emission factor. Unfortunately, the errors in the CO and $CO_2$ emission factors are not correlated, limiting the use of CO in constraining $CO_2$ at the national scale []." (lines 83-86).

*L. 82: Please add "relative" before "error", i.e. "So that the relative error in CO2 emissions…".*

Done.

*Methods:*

*Throughout: There are several abbreviations, which are not spelled out in full, e.g. "TNO", "CAMS-REG", "CEIP", "LRTAP", "UNFCCC", "STEAM", "IPCC", "EMEP", "VISUM", "GMAO", "CASA-GFED". I think the EGU guidelines require to spell out the full names for such abbreviations.*

We appreciate the suggestion from the reviewer. We have written out the abbreviations where deemed necessary. For some abbreviation, such as UNFCCC and IPCC, we believe the abbreviations are well-known and the full names are less familiar to the readers.

*L. 155-157: "Where needed" seems to be a bit vague. Can you give some illustrative examples, e.g. for which situations you selected the largest uncertainties from a given range? This can also be done in the supplements.*

This is an error on our side. We have corrected this with the following statement: "When multiple fuel types are used within a sector we pick the dominant fuel type." (lines 169-170).

*L. 169: Maybe you could even say "…, which are the most important contributors to CO and CO2 emissions from area sources…".*

We have included the suggested changes (line 182).

*L. 172-173: I didn't get this sentence. Are there also countries for which emissions are not downscaled by using density population or traffic volume proxy maps? Please clarify.*

We only apply our own downscaling for countries with reported emissions. For other countries on the borders of our domain we add gridded emissions from other datasets, as mentioned earlier in the manuscript. We have added a clarification (lines 186-187).

*L. 195: Please explain what "categorical proxies" are.*

Categorial proxies are proxies that are based on the presence of certain characteristics (e.g. land use types) instead of having a numerical value. We have added this explanation (lines 212-213).

*L. 220ff: It might be useful to briefly introduce the concept of semi-variograms here, as you derive from those the spatial error correlation lengths.*

We have added two sentences describing the concept of semi-variograms: "A semi-variogram describes the spatial autocorrelation as a function of distance, i.e., the degree of variability between points located at a certain distance from each other. In the case of the proxy maps points that are closer together are expected to be more similar, and therefore their errors are more strongly correlated." (lines 239-242).

*L. 246-247: Do you mean "… the relative contribution of each proxy map … "? Also, what does the vector WD mean? Does the vector WD require subscripts? Do you calculate in Eq. 3 the standard deviation and the maximum of the different WD values of the contributing proxy maps in the respective grid cells? Please clarify.*

The vector WD indeed needs subscripts $c$ and $m$, this is now corrected. This makes it clear that the standard deviation and maximum value calculated in Eq. 3 are per grid cell. Moreover, a few clarifications have been added in the text, e.g. on when we are referring to relative and absolute uncertainties (lines 265-269).

*L. 254: The subscript "C" of "P_C" in Eq. 3 can easily be mixed up with "c" for grid cell. Please clearly indicate that "P_C" and "P_F" are the predictors for the GNFR C and GNFR F sectors. More importantly, why do you need different predictor definitions for the two different sectors?*

We have renamed the predictors to $PC$ and $PF$ for GNFR C and GNFR F, respectively, to avoid confusion with the subscript $c$ for the grid cells. The reason that the predictors are slightly different is that the sectors behave differently due to their respective characteristics. For example, for GNFR F the number of proxy maps is often larger and by taking the standard deviation we lose of a lot of predictive power, whereas for GNFR C the standard deviation had the best match with the correlation coefficients following from the Monte Carlo approach.

*L. 255: It may be useful to add here a brief description on how the Monte-Carlo approach was applied to calculate the correlation coefficients.*

We have added the following explanation: "For the Monte-Carlo method an ensemble of gridded emissions was produced by randomly perturbing the grid cell emissions of $CO_2$ and CO for each proxy map following the defined uncertainty ranges. The perturbations are applied equally to $CO_2$ and CO emissions, assuming full error correlation for each proxy map, which is a valid assumption at the grid cell level. The correlation coefficient results from a linear regression on the total $CO_2$ and CO emissions per grid cell for a given GNFR sector." (lines 280-284).

*L. 292ff: It would be helpful if you could split Eq. 6 into two (or even three) equations: one equation for the aggregation of the sub-sector emission uncertainty, one equation for the calculation of the grid cell relative uncertainty shown in Fig. 3, and one equation for the calculation of the weighted average correlation length. Then you could also refer to the respective equations in Fig. 2 more properly.*

We have added a new equation 7, which is the same as equation 6 except for indices of the standard deviation (line 328). This equation is related to the grid cell uncertainties. This has also been updated in Fig. 2. For the correlation length there is no uncertainty propagation involved, as the correlation length itself is not an uncertainty. Here, a simple averaging over all sub-sectors is performed, as shown in Fig. 2.

*L. 323-324: Do you use the same transport model (GEOS-Chem) for the global simulation? And what is the temporal resolution of the anthropogenic CO2 and CO emissions, i.e. do you apply the diurnal time profiles from TNO?*

Yes, GEOS-Chem is also used for the global simulations (line 356). The temporal resolution is daily, using the monthly and weekly profiles from TNO. This information has been added (lines 357-358).

*L. 373: Are you using hourly CO2 and CO observations? Also, considering the goals of the World Meteorological Organization (WMO) for the in-situ CO2 measurement uncertainty of 0.1ppm, I would expect the 2ppm CO2 uncertainty as an upper limit. Maybe you can state this.*

The observations are 3-hourly averages to match the temporal resolution of the model's meteorology. We only use observations between 0900 and 1800 local time. This information has been added (lines 408-409). The observation uncertainty is a combination of errors related to the instrumentation and errors in the representation of the observations by the transport model (e.g. when comparing grid cell average concentrations to point observations) (line 408). Therefore, it is higher than just the instrumentation error. The model uncertainty refers to errors in the model parameterization and model input.

*Results:*

*Fig. 5: It would be easier to compare the different panels, if the two panels on the left (and right) side have the same x-axis.*

Both the x-axis and the bin size have been made identical for all panels.

*L. 413: This sentence is unclear to me. Could you please explain this with a bit more detail. Maybe you could also refer to the respective equations in the methods and to their parameters (e.g., by "standard deviation", do you mean "stdev(WD)" in Eq. 5?). This would again allow a better understanding of the predictor definition.*

We have tried to clarify this statement, including a reference to the Eq. 3 (lines 452-453).

*L. 439-440: Do you mean: |prior - true| - |posterior - true|? Maybe you could provide an equation for this metric shown in Fig. 8 (could also be done in the figure caption).*

An equation has been added (lines 480-481).

*Fig. 9: Maybe you could write the mean values of the error correlations into the respective panels. Are the differences in the mean error correlations of the four experiments significant? Is there a reason, why you have chosen a larger bin-size for the histograms in the upper panels than in the lower panels? Please also correct the caption: "...for four experiments".*

The mean values have been added to the panels, the bin sizes are made equal for all panels and the caption is corrected. Although the differences are small between the experiments, we see more values in the bins around zero, indicating more ability to differentiate between fossil and biogenic fluxes of CO2. We have added a clarification (lines 501-502).

*Fig. 10: Please explain the dashed line in the caption. How would these plots look like for the Adv_CO2 and Adv_CO2_CO experiments?*

The dashed line is the 1:1 line, which is now mentioned in the caption. The results for the Adv_CO2 and Adv_CO2_CO experiments are not significantly different, as most of the information already goes into updating the biogenic fluxes due to its high uncertainty. This explanation has been added as well (lines 521-522).

*Discussion and conclusions:*

*L. 557-561: Can you see seasonal differences in the performance of CO to constrain fossil fuel CO2 emissions? I wonder whether CO can provide more or less additional constraint during winter (with heating and traffic*

*emissions in Central Europe) compared to during summer (without heating emissions). Are there seasonal (winter vs. summer) differences in the posterior error correlation between biogenic and fossil fuel CO2?*

Yes, there are seasonal differences in the performance of CO as a constraint to fossil fuel CO2. During summer months the biogenic signals are much stronger and the CO emissions are lower, making CO a poorer constraint on CO2 fossil fuel fluxes. Although the corrections on CO2ff are larger in summer, they are more often moving away from the truth, especially in areas where the errors in biogenic and fossil fuel CO2 are opposite. This is also expressed by stronger positive posterior error correlations between the biogenic and fossil fuel CO2 components, indicating that both are corrected in the same direction, despite having opposite prior errors. See below figures for May and November (Adv_CO2_CO). In November the posterior error correlations are weaker and more often negative. We also see that during winter CO generally improves the corrections on CO2ff. Each month may have its own challenges and based on one year of monthly averaged data it is hard to draw conclusions on this. For example, we also see winter months in which some areas perform worse and summer months in which some areas are much better constrained. We have added a few sentences on the seasonal differences, but with the notion that we cannot draw definite conclusions yet (lines 504-508).

[Figure]

*L. 569-570: Please emphasize that the used in-situ observations are synthetic data.*

Done.

*L. 570ff: Do you have an explanation for why the fluxes in France are hardly altered, although there are some French observation sites? Are those sites perhaps less influenced by fossil emissions? More generally, could the low additional benefit of CO in the Adv_CO2_CO inversion partly be explained by the fact that some of the CO2&CO observation sites are less influenced by fossil emissions? In other words, could CO2 & CO observations from urban sites lead to a greater benefit from the Adv_CO2_CO experiment?*

The ICOS atmospheric stations are indeed set up in remote areas with little local influence. Hence, the biogenic fluxes dominate the measured signals and it is less costly to assign a model-data mismatch to the biogenic fluxes than to fossil fuel fluxes. Additionally, we plot absolute changes in emissions and therefore areas with low emissions are unlikely to show in the plots, even if they receive relatively large corrections. Therefore, we mostly see corrections in fossil fuel fluxes in areas with strong human activities that are within reasonable distance from an observation site. Therefore, Paris is visible in the plots and the surrounding area is not, as an example.

It would definitely help to have more observation sites close to human/industrial activities for constraining fossil fuel fluxes. The larger the relative share of the fossil fuel signal compared to the biogenic signal, the easier it is to make corrections on it. We have added some discussion on this (lines 625-630).

*Technical corrections:*

*L. 159: Add a blank space in the reference "(... August 25, 2022)"*

Done.

*L. 194: "difficult uncertainty"*

Done.

*L. 199-200: "an uncertainty"*

Done.

*L. 210: Add a blank space in the reference "(... September 15, 2022)"*

Done.

*Table 3: Maybe you could slightly change the format of the title column in the table, because there is a lot of space between "CO:CO2" and "error", which is why it looks as "error" would be a fifth column.*

Done.

*L. 446: "2000 kg s-1"*

Done.

*Supplement material:*

*L. 16 & L. 25: What is "AP"? Do you mean "AD"?*

AP is short for air pollutants. We have written this out in full in both instances.

*L. 89-90: Do you mean "simple uncertainty propagation method"?*

Indeed, this has been corrected.

***Anonymous Referee #2, 13 Jul 2024***

*The paper describes a method to estimate CO and CO$_2$ emission uncertainties and their covariances, including cross-tracer error covariances, which is the main novelty of the approach. These emission uncertainties are then used in an inverse modelling experiment, with a synthetic truth. I found the first part of the paper rather robust and convincing, and actually very relevant for the emission estimation communities (bottom-up and inverse). However, I have more reserves regarding the inverse modelling part: these inverse modeling experiments are constructed to demonstrate the interest of using the emission error correlations constructed in the bottom-up part, but I didn't find the demonstration very convincing (see major comment below). Despite this reserve, the paper is in a very good shape and quite interesting already, I think the critique can be addressed through a minor revision.*

We thank the reviewer for this positive note. We wanted to demonstrate the value of the emission uncertainties also as a way of validation and it is great to hear that the emission uncertainties in itself are already valued.

*Major comment*

*The major issue I have with the study is that the inverse modeling section is constructed around a set of experiments with a synthetic truth, designed to test the capacity of the inversions to recover the emissions, given different type of emission uncertainties. In both case, a set of "true" emissions is defined, corresponding "true" observations are generated, by forward propagation through GEOS-CHEM. A set "prior" emissions is generated by applying a perturbation to that truth, and from there, two sets of inversions are performed:*

*In the "Base" cases, errors on CO and CO2 emissions are assumed to be fully correlated*

*In the "Adv" cases, error correlations are inferred from the bottom-up modelling section*

*The issue is that the perturbation is essentially generated as a realisation of the multi-variate PDF represented by the emission error covariance matrix of the "Adv" inversions (Eq. 10). So it's only logical that "Adv" performs*

*better. If the perturbation had been generated based on the "Base" error covariances, then "Base" would perform better.*

*I agree that the covariance matrix of "Adv" is probably a more realistic covariance matrix than the one in "Base", and it is still interesting to see the type of impact that this has on the results, but it is not a very strong demonstration.*

This is a valuable comment from the reviewer. We agree that the results are biased towards the "Adv" covariance matrix, as we use this as our best guess of the true relationship between $CO_2$ and CO errors. Basing the "truth" on something we know for sure isn't correct (i.e., a 100 % error correlation between $CO_2$ and CO, as in the "Base" case) wouldn't make a lot of sense. However, previous studies assumed a full error correlation for $CO_2$ and CO fossil fuel fluxes. Therefore, we aim to show that this faulty assumption may have a large impact on the results. In other words, we test whether adjusting the $CO_2$ and CO error statistics to be closer to the "truth" would provide a benefit over the "Base" scenario (lines 414-416).

Whether or not the "Adv" prior error definition is a correct representation of the truth we cannot know for sure, but at least we are confident that it is more realistic than the "Base" case. Moreover, we add significant random noise to the "observations" and with the "Base" and "Adv" errors being of similar order of magnitude we believe to have a realistic comparison of the experiments. Further tests can be done to determine whether the "Adv" error statistics still provide a benefit over "Base" if the true relationship between $CO_2$ and CO is different than our assumption in these experiments, but that warrants a whole separate study.

*Minor comments*

*Throughout the paper, the authors refer to GNFR A, B, C, D, etc. This is quite hard to follow for readers not used to that terminology, as it obliges to refer to the table every time. It would make the reading a lot easier to give the category name whenever possible, in addition of the GNFR code. This is done in a few instances (l168), but not enough.*

We have included the category name throughout the manuscript.

*l232-234: I don't really understand the rationale here: even though these are point sources, aren't the errors in the emission factors correlated?*

The spatial correlation length, which is discussed here, describes the correlation in the spatial uncertainties. As described in Section 2.1.3 the spatial uncertainty takes into account the value of the spatial proxy itself and how representative it is for the emission sector. In the case of point sources we mostly know the exact location, so the spatial uncertainty is non-existent. Hence, there is no spatial error correlation length. This has been added to lines 252-253. Indeed, the emission factors uncertainties may be correlated, but this is expressed in the national total emission uncertainty, as it will affect the total emissions in a country (and therefore the emissions at each point source) and not the location of the emissions.

*l247: what does "WD" stand for?*

"WD" stands for weighted difference, which is a vector describing the absolute weighted difference between the CO2 and CO emissions per sub-sector (line 269).

*l273: "the standard deviation of a Gaussian distribution belonging to a lognormal distribution" ==> that sentence is not clear to me*

The simple error propagation equations make use of standard deviations of Gaussian distributions. Therefore, we need to transform the lognormal error distributions and estimate the standard deviation of their equivalent Gaussian distributions. We have clarified this in lines 301-303.

*l409: If I read the sentence literally, it means you have a correlation coefficient > 1 ...*

We thank the reviewer for noting this. It is correct that for some countries the fitted function has a maximum value of slightly more than one, suggesting a correlation coefficient larger than 1 is possible. This has no physical meaning and we simply set the maximum value to one (lines 447-448).

*Figure 6: maybe use hexbin instead of scatter for these plots (or use some transparency, or make your dots smaller ...), as it's unclear where the density of dots is the highest when they all overlap*

The scatter plots have been replaced with hexbin plots.

*Figure 7: it would be better to have all subplots on the same y-range*

We politely disagree with the reviewer, since both the x- and y-range differ strongly and making them all equal will show almost no data in a few graphs. However, we do add a notice that the axes are different (lines 474-475).